# Contraction Property of Pooling Layer

## Abstract

Although the theory of deep neural networks has been studied for years, the mechanism of pooling layers is still elusive. In this paper, we report the angle contraction behavior of pooling strategies (the average pooling and max pooling) at initialization. Compared to the relu-activated fully connected layer or convolutional layer, the pooling layer stands as the main source of contraction of the angle between hidden features. Moreover, we show that the cosine similarity between average pooling features in convolutional neural network is more data-dependent than fully connected network, while the max pooling is not sensitive to the data distribution in both architectures. Our results may complement the understanding of the representation learning.

## 1 Introduction

The idea of stacking many layers to make truly deep neural networks (DNNs) is what arguably led to the neural net revolution in the 2010s. Indeed, from a function-space point of view, it is known that depth exponentially improves expressivity Poole et al. (2016); Eldan & Shamir (2016). However, a noteworthy fact is that under standard initialization deep neural networks become increasingly degenerate as depth gets larger. One type of degeneracy is known as vanishing/exploding gradients He et al. (2015b); Hanin (2018). Another type of degeneracy is that a neural network contracts all features to restricted directions, and we call it angle contraction in this paper. Cho et al. were the first to compute the relu-activated fully-connected neural network, showing that it acts as an arccosine kernel Cho & Saul (2009): $\hat{r}(\rho) = \frac{\sqrt{1-\rho^2}+\left(\pi-\cos^{-1}(\rho)\right)\rho}{\pi}$ for the cosine similarity $\rho(x,y) = \frac{\langle x,y \rangle}{\|x\|\|y\|} \in [-1,1]$ between tensors $x$ and $y$ (the notations and terms are presented in section 1.3). The kernel will contract the angle of the features. This phenomenon later has been reported and inspected by several researchers Avelin & Karlsson (2022); Schoenholz et al. (2017); Yang & Schoenholz (2017); Hayou et al. (2019); Nachum et al. (2022). which is also termed dual activation of relu network Daniely et al. (2016). Hayou et al. studied the arccosine kernel of the NNGP of ResNet Hayou et al. (2021). Martens et al. dealt with the degeneracy phenomenon in the approach of activation function shaping Martens et al. (2021). Nachum et al. Nachum et al. (2022) reported that for convolutional neural networks, the extent of degeneracy was dependent on the type of input. A detailed analysis of the angle contraction between features for deep ReLU network is performed by C. Jakub and M. Nica Jakub & Nica (2024).

### 1.1 Contribution

The relu activation is not the only source of angle contraction. In practice, the well-known CV models such as VGG, Resnet, and DenseNet can ue batch normalization to avoid the contraction. However, we still notice the angle contraction phenomenon at the penultimate layer, i.e., the layer before the final classifier layer, the main reason of which is that all the models use global average pooling before the feature is fed to the classifier. The average pooling contracts points heavier than a fully-connected layer or a convolutional layer and stands as the main source of the the angle contraction at initialization. Other than average pooling, the other popular pooling strategy, the max pooling, also contracts points severely. In contrast to Nachum et al. (2022), we find CNN without average pooling fits the usual arccosine kernel for several types of data (Gaussian, Cifar10, Cifar100, and MNIST) which is data-independent, while CNN with average pooling induces data-dependent contraction.

The contribution of this paper is as follows:

1. we report the angle contraction phenomenon caused by pooling strategy; 2. we show theoretically why the angle contraction of average pooling is not data-dependent when pooling strategies are applied to fully connected network. 3. We show theoretically that angle contraction is not data-dependent for max pooling in both FCN and CNN 4. we find empirically that average pooling in CNN results in a data-dependent angle contraction. 5. we explain why average pooling features of the Gaussian images experience severe contraction.

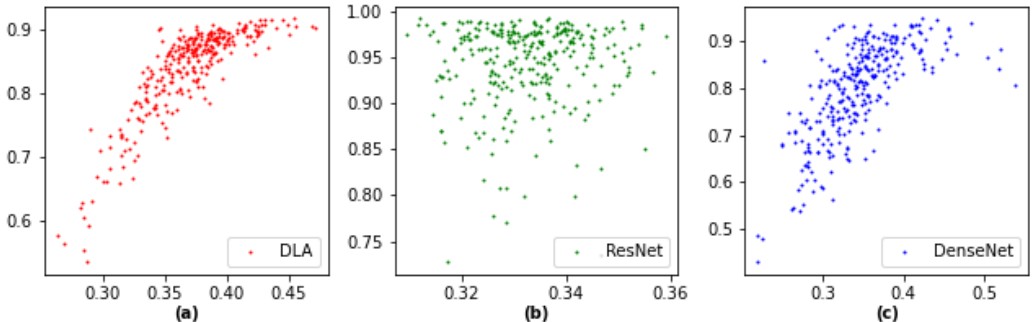

Figure 1: Angle contraction of random pairs in mean-shifted cifar10 dataset through DLA, ResNet32, and DenseNet150. The horizontal axis is the cosine similarity of relu-activated features, and the vertical axis is the cosine similarity of average pooling features.

## 1.2 RELATED WORK

There are several mainstream pooling strategies and they have a lot of applications in deep neural networks. The most relevant are average pooling and max pooling.

The average pooling was initially introduced in LeCun et al. (1989) and used in the first convolutional neural network LeCun et al. (1998). An average pooling layer partitions the input into square (or rectangular) regions and computes the average value of each region. lin et al. proposed Global Average Pooling (GAP) to aggregate the feature maps in the classifier Lin et al. (2014). Popular deep learning models such as GoogLeNet Szegedy et al. (2014), DenseNet Huang et al. (2017), ResNet He et al. (2015a), and DLA Yu et al. (2018) use GAP by default.

Max pooling was introduced in Ranzato et al. (2007) to learn sparse representations for deep belief network. It partitions the input into square (or rectangular) regions and computes the maximum value of each region. it has been applied to myriads of models for classification and segmentation including AlexNet Krizhevsky et al. (2012), VGG Simonyan & Zisserman (2015), U-NetRonneberger et al. (2015), and YoloRedmon et al. (2016).

Angle contraction phenomenon is mostly studied for MLP and CNN from the perspective of Neural Tangent Kernel Jacot et al. (2018). Arora et al. (2019), and Bietti and Mairal (2019) give expresvsions for the NTK and the convolutional NTK (CNTK). Arora et al. (2019) provide concentration bounds for the NTK of fully connected networks with finite width. Bietti and Mairal (2019) derive smoothness properties for NTKs, e.g., upper bounds on the deformation induced by the NTK in terms of the initial Euclidean distance between the inputs. A related approach to NTK is taken in (Bietti, 2021) where convolutional kernel networks (CKN) are used.

at the time of the submission, we are not aware of literature that studies the contraction phenomenon from the perspective of pooling layer.

## 1.3 NOTATION

$\mathbb{R}$        The set of real numbers.

$d$        the size of image

| | |
|---|---|
| $\mathcal{D}$ | The set of double index $\{(1,1),(1,2),\ldots,(d,d)\}$ |
| $[n], [m,n]$ | The set $\{1,2,\ldots,n\}$ and the set$\{m, m+1, \ldots, n\}$ for $m \le n$, resp. |
| $a_i \ A_{ij}$ | The $i$'th entry of vector a, and the $(i,j)$ entry of the matrix $A$, resp. |
| $\alpha_i, \beta_j$ | The patch $i$ of 2D tensor x and the patch $j$ of 2D tensor y, where $i,j \in \mathcal{D}$, the ordered set of coordinates of the pixels in images $x$ and $y$ of size $d \times d$. |
| $I_k$ | The identity matrix of dimensions $k \times k$. |
| $\text{vec}(\cdot)$ | The vectorization operation of a matrix or a tensor in some systemic order. |
| $\langle x, y \rangle$ | The standard inner product between vectors $x$ and $y$. For matrix or tensor $x$ and $y$ (of the same dimensions) the inner product is defined as $\langle x, y \rangle = \langle \text{vec}(x), \text{vec}(y) \rangle$. |
| $\|x\|$ | The standard Euclidean norm, induced by the standard inner product: $\|x\| = \sqrt{\langle x, x \rangle}$. |
| $\rho(x,y)$ | The (cosine) similarity. The similarity between $x$ and $y$ is defined as $\rho = \frac{\langle x,y \rangle}{\|x\|\|y\|}$. |
| $\bar{\rho}(X,Y)$ | The mean similarity. $\bar{\rho}$ between random $X$ and $Y$ is defined as $\bar{\rho} = \frac{\mathbb{E}\langle X,Y \rangle}{\sqrt{\mathbb{E}[\|X\|^2]\mathbb{E}[\|Y\|^2]}}$. |
| $r(x)$ | The ReLU applied to $x$ by $r(x) := \max\{0,x\}$; for a tensor $x$ the $r$ is applied entrywise. |
| $\hat{r}(\rho)$ | The dual activation of the ReLU at $\rho \in [-1,1]$: $\hat{r}(\rho) := \frac{\sqrt{1-\rho^2}+(\pi-\cos^{-1}(\rho))\rho}{\pi}$. |
| $\mathcal{N}(\mu, C)$ | A Gaussian vector with mean vector $\mu$ and covariance matrix $C$. |
| $ind(\alpha)$ | index of the entries of patch $\alpha$ of a 2D tensor (the 2D image). |
| $\mathcal{A}, \mathcal{M}$ | average pooling operator, max pooling operator, average over discrete multiset $S$ or all entries in a 2D tensor, resp. |
| $\max(\cdot)$ | the maximum value of a set or the maximum pixel value of a 2D tensor. |
| $Ave(\cdot)$ | average over discrete multiset $S$ or all entries in a 2D tensor, resp. |

## 2 THE PROBLEM SETTING

The networks we study are 1-hidden layer relu fully connected network (abbr. FCN) and 1-hidden layer relu convolutional neural network (abbr. CNN).

Let $d_{in}$ and $d_{out}$ be two positive integers, FCN is defined as

$$F(x) = r(Wx) \tag{1}$$

where $x \in \mathbb{R}^{d_{in}}$ and $W$ is a $d_{out} \times d_{in}$ matrix. we call $F(x)$ the **relu-activated feature**. Let $d_{out} = mn$ and the index of hidden neurons be partitioned into m components, say the partition $\mathcal{P} = \{P_1, P_2, \ldots, P_m\}$ with $P_i = \{in+1, \ldots, (i+1)n\}$ for $i \in [m]$. Then the pooling operator $\mathcal{A} : \mathbb{R}^{mn} \to \mathbb{R}^m$ for FCN is defined as:

$$\mathcal{A}(F(x)) := (\mathcal{A}(F(x))_1, \mathcal{A}(F(x))_2, \mathcal{A}(F(x))_m) \ with \tag{2}$$
$$\mathcal{A}(F(x))_i = Ave(\{x_i\}_{i \in P_i}) \quad \forall \, i \in [m]. \tag{3}$$

we call $\mathcal{A}(F(x))$ the **average pooling feature**. Similarly, we can define $\mathcal{M} : \mathbb{R}^{mn} \to \mathbb{R}^m$ over the same space as

$$\mathcal{M}(x) = (\mathcal{M}(x)_1, \mathcal{M}(x)_2, \mathcal{M}(x)_m) \ with \tag{4}$$
$$\mathcal{M}(x)_i = \max(\{x_i\}_{i \in P_i}) \quad \forall \, i \in [m]. \tag{5}$$

$\mathcal{M}(F(x))$ is termed **max pooling feature**.

The output $z \in \mathbb{R}^{d \times d}$ of a single convolutional kernel is given by

$$z = r(W * x) \tag{6}$$

where $x \in \mathbb{R}^{d \times d}$ is the input, $W \in \mathbb{R}^{k \times k}$ is the kernel, and $k$ is kernel size of $W$. The two dimensions (both are $d$ in our setting) of $x$ and $z$ are the image dimensions (height and width), and in this paper, we only consider 1-channel images theoretically. For convenience, we assumes $k$ is odd. the $*$ operation is the convolution

$$(W * x)_{uv} = \sum_{i,j \in [-\frac{k-1}{2}, \frac{k-1}{2}]} W_{ij} x_{u-i, v-j}.$$

Then the CNN is defined as $F : \mathbb{R}^{d \times d} \to \mathbb{R}^{d \times d \times m}$ by:

$$F(x) = (r(W^1 * x), r(W^2 * x), \ldots, r(W^m * x)) \tag{7}$$

where $m$ is the number of channels. We abuse the symbols $W$, $F$, and $m$ Since the two cases since there is no ambiguity. Now we can define the global average pooling feature of the relu-activated CNN feature $F(x)$:

$$\mathcal{A}(F(x)) := (Ave(r(W^1 * x)), Ave(r(W^2 * x)), Ave(r(W^m * x))) \tag{8}$$

Similarly the (global) max pooling feature of the relu-activated CNN feature $F(x)$ is defined as

$$\mathcal{M}(F(x)) := (\max(r(W^1 * x)), \max(r(W^2 * x)), \max(r(W^m * x))) \tag{9}$$

For simplicity, denote $\mathcal{A}_x := \mathcal{A}(F(x))$ and $\mathcal{M}_x := \mathcal{M}(F(x))$ in the following content.

# 3 EMPIRICAL EVIDENCE

This section presents the empirical verification of the angle contraction of pooling layers. The architectures we use in the experiments are 1-hidden layer relu FCN and 1-hidden layer relu CNN. For FCN, the width of the hidden layer is 1000; For CNN, the output channel is 640. Both models use standard He initialization He et al. (2015b). The pooling layer for CNN is a global pooling, and that for FCN is a 1D average pooling with kernel size = 50.

## 3.1 ARE THE POOLINGS DATA-DEPENDENT?

The answer is: It depends. We show that for fully connected layer, Figure 3.1 shows that both pooling strategies result in a heavy contraction for a fully connected layer, where all the cosine similarity values are nearly 1; Figure 3.1 shows for convolutional layer, average pooling has a slighter contraction than the max pooling. These observations conform to the theoretical analysis.

It is noticeable that max pooling contracts points heavily for both FCN and CNN whatever the data distribution is; On the other hand, the average pooling induces a mild contraction on natural datasets cifar10 and cifar100, but a severe contraction on gaussian noise images and artificial images from MNIST.

To explore the variability of the average pooling we construct a toy data pair $x = \begin{pmatrix} \mathbf{1}_{\kappa \times \kappa} & -\mathbf{1}_{\kappa \times (d-\kappa)} \\ -\mathbf{1}_{(d-\kappa) \times \kappa} & -\mathbf{1}_{\kappa \times \kappa} \end{pmatrix}$, and $y = \begin{pmatrix} -\mathbf{1}_{\kappa \times \kappa} & \mathbf{1}_{\kappa \times (d-\kappa)} \\ \mathbf{1}_{(d-\kappa) \times \kappa} & \mathbf{1}_{\kappa \times \kappa} \end{pmatrix}$, which have zero sample mean and unit sample variance. Then Table 2 shows how the the cosine similarity of two average pooling features (the penultimate column of Table 2) changes with the parameter $\kappa$, making it difficult to make a general conclusion about the angle contraction of global average pooling in CNN. On the contrary, the cosine similarity of the max pooling features (the rightmost column of Table 2) or relu-activated features (the middle column of table 2) is not affected easily by $\kappa$, indicating the insensitivity of the max pooling (or relu activation) to the data.

| $\kappa$ | $\rho(x,y)$ | $\rho(F(x), r(F * y))$ | $\rho(\mathcal{A}(F(x)), \mathcal{A}(F(x)))$ | $\rho(\mathcal{M}(F(x)), \mathcal{M}(F(x))$ |
|---|---|---|---|---|
| 5 | -0.9922 | 0.0002 | 0.0187 | 0.7247 |
| 10 | -0.9684 | 0.0006 | 0.1030 | 0.6787 |
| 15 | -0.9177 | 0.0012 | 0.3587 | 0.7176 |
| 20 | -0.8696 | 0.0017 | 0.6619 | 0.7256 |
| 25 | -0.7952 | 0.0022 | 0.9771 | 0.7174 |

Table 2: Variability of the cosine similarity of pooling features

# 4 THEORETICAL ANALYSIS OF THE CONTRACTION

In this section, we theoretically analyze the contraction behavior of the average pooling features. The contraction in FCN is data-independent in theory, and the contraction for CNN only works for Gaussian data due to the complex correlation between the entries of the pooling features.

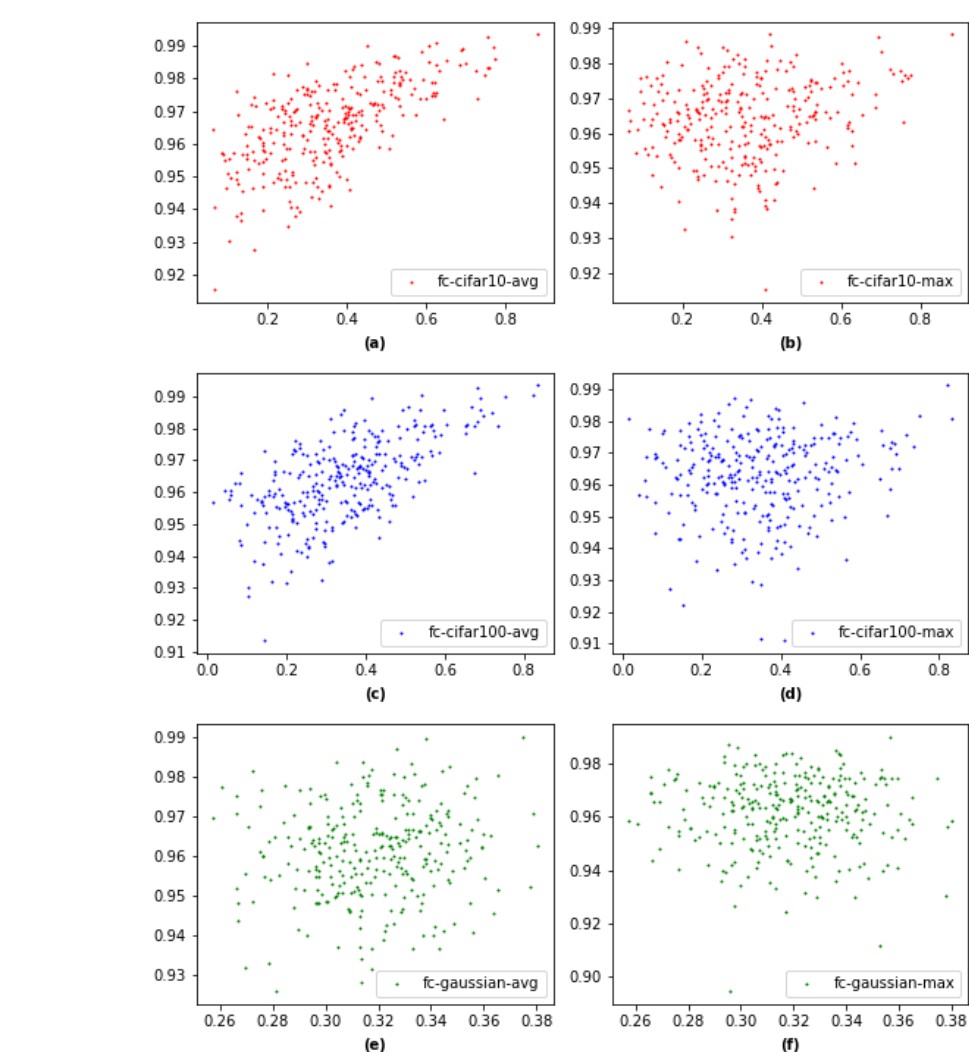

Figure 2: Cosine similarity of pooling features (vertical axis) vs cosine similarity of relu-activated features (horizontal axis) in fully-connected network for centered cifar10, centered cifar100, and gaussian data.

**Theorem 4.1.** *For FCN, let the entries of $W$ be initiated as i.i.d standard normal random variables. Then for any pair of vectors $x$ and $y$, $1 - \rho(\mathcal{A}_x, \mathcal{A}_y) = O(n^{-1})$*

The theorem is easy to understand; it indicates that as long as $n$ (kernel size) is large, the angle between average pooling features is close to 0.

The same technique is applicable to the max pooling strategy:

**Theorem 4.2.** *For FCN, let the entries of $W$ be initiated as i.i.d standard normal random variables. Then for any pair of vectors $x$ and $y$, $1 - \rho(\mathcal{M}_x, \mathcal{M}_y) = O(log^{-1}n)$*

It is surprising that max pooling in CNN has almost identical contraction behavior to that in FCN. This is explained in the following corollary.

**Corollary 4.3.** *For CNN, let the entries of $W$ be initiated as i.i.d standard normal random variables. Then for any pair of vectors $x$ and $y$, $1 - \rho(\mathcal{M}_x, \mathcal{M}_y) = O(log^{-1}n)$*

The corollary holds true since the variance is very small after rescaling (see the proof in the appendix), but this argument does not hold for average pooling since the standard deviation is the

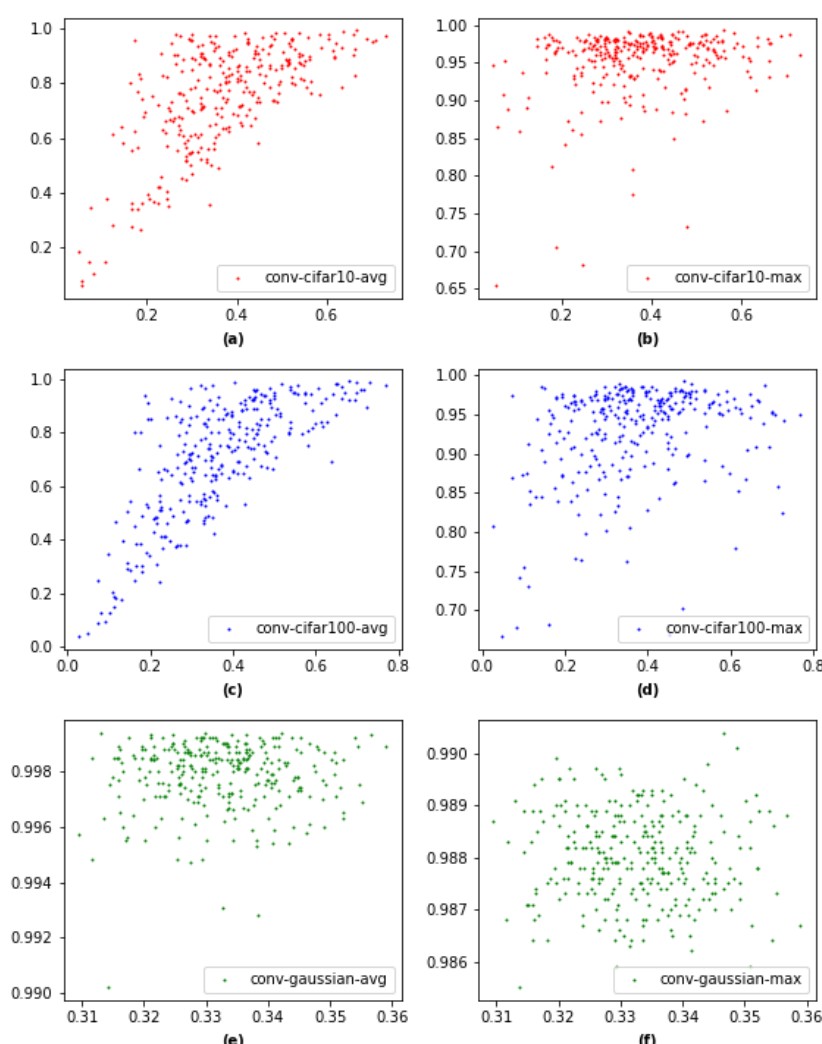

Figure 3: Angle contraction of poolings after convolutional layer for cifar10, cifar100, and gaussian data.

same order of magnitude as the mean (see the discussion in the appendix D). We realize that for average pooling features, the contraction behavior may be more complicated. In the following theorem, we prove that the contraction behavior is chaotic for Gaussian data, i.e., only depends on the size of the image. Let $\mathcal{C} := \frac{\mathbb{E}[\mathcal{A}r(x)\mathcal{A}r(y)]}{(\mathbb{E}[(\mathcal{A}r(x))^2]\mathbb{E}[(\mathcal{A}r(y))^2])^{1/2}}$, where the expectation is taken over all the randomness, then we have the following theorem:

**Theorem 4.4.** *For $x$ and $y$ whose entries are i.i.d from $\mathbf{N}(0,1)$, and whose joint distribution is $(x, y) \sim \mathbf{N}(0, I_{d^2})$ then $\rho(x, y) \approx 0$, $\rho(r(x), r(y)) \approx \frac{1}{\pi}$, and $\mathcal{C} = 1 - O(d)$.*

## 5 DISCUSSION AND LIMITATION

Nachum et al. (Nachum et al. 2022) show that 1-hidden layer CNN is a nearly isometrical map on the natural image data such as cifar10, cifar100, and ImageNet. We show empirically that this phenomenon is the consequence of the non-centering of the data rather than the naturality of the data. We normalize cifar10 dataset by its sample mean and standard deviation and shift them by a constant. We randomly pick 300 pairs of images, feed them into the CNN, and then calculate the cosine similarity values of all the pairs. Figure 5 shows the angle contraction is not affected

by a small mean shift (Figure5(a)-(c)); as the shift increases, the degree of contraction becomes slower than the arccosine kernel (Figure5 (d)-(e)). we observe that at mean = 2.0 (Figure5(e)), the contraction behavior mimics the original non-centered cifar10 data (Figure5 (f)). On the other hand, the average pooling in CNN is data-dependent.

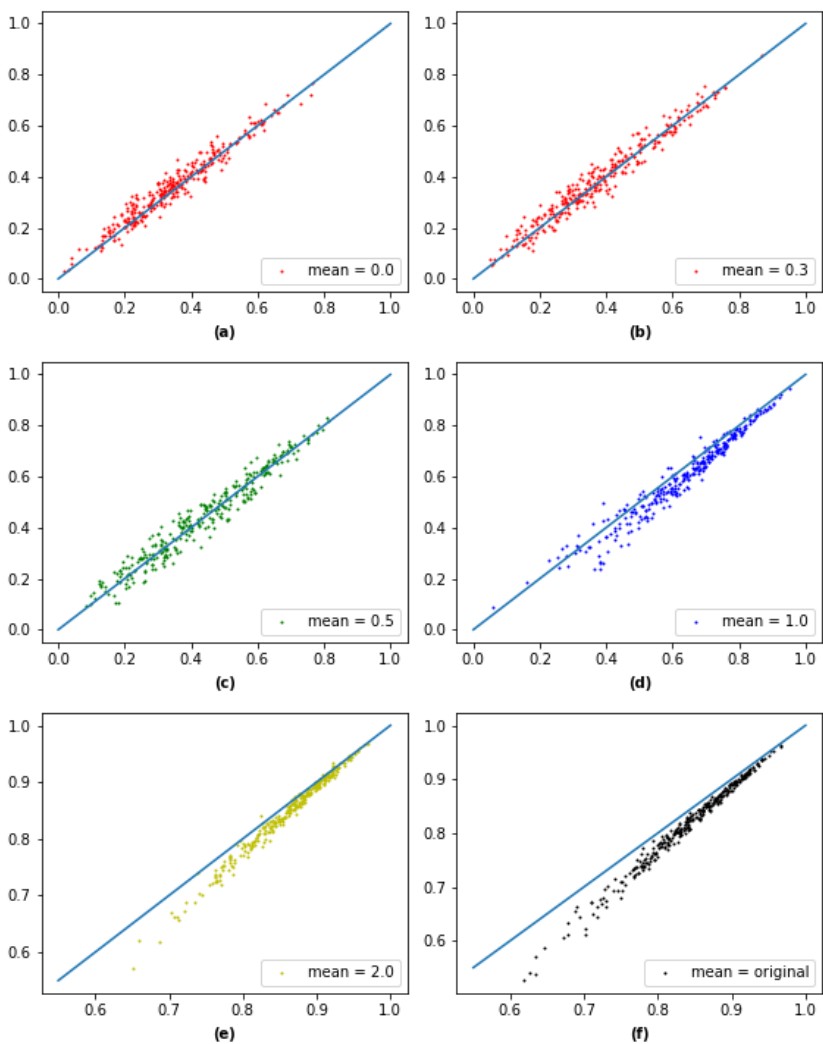

Figure 4: Angle contraction of random pairs in mean-shifted cifar10 dataset through the 1-hidden layer CNN. The horizontal axis is the cosine similarity of the relu-activated features and the vertical axis is the arccosine kernel $\hat{r}(\rho)$ of the cosine similarity of the input data. The label shows the extent of the mean shift.

We are not able to give a general answer to the concentration behavior of the average pooling in CNN. We have constructed examples to show that cosine similarity between average pooling features has a high variability compared to that in FCN, we hope to obtain a more thorough theoretical justification in future work.

According to the experiments, the theoretical bound for max pooling seems not optimal since the maxpooling has heavier concentration than average pooling in both architectures. The reason behind the observation may deserve inspection.

It remains unclear how the contraction behavior of average pooling at initialization connects to the performance of the specific architectures. Combining Figure 1.1 and Figure 5, we can see that DLA, DenseNet150 and ResNet50 have increasing contraction behavior, that is, similar relu-activated fea-

tures have similar average pooling features in the sense of expectation of all the random pairs, and vise versa; ResNet32, however, does not follow this behavior and induces chaotic cosine similarity. DLA, DenseNet150, ResNet50, and ResNet32 achieves 96.4%, 95.8%, 95.2%, 93.5% test accuracy on cifar10, resp. Can we measure the generalization of the learning model by checking its contraction behavior at initialization? This will be left as an open problem for the future.

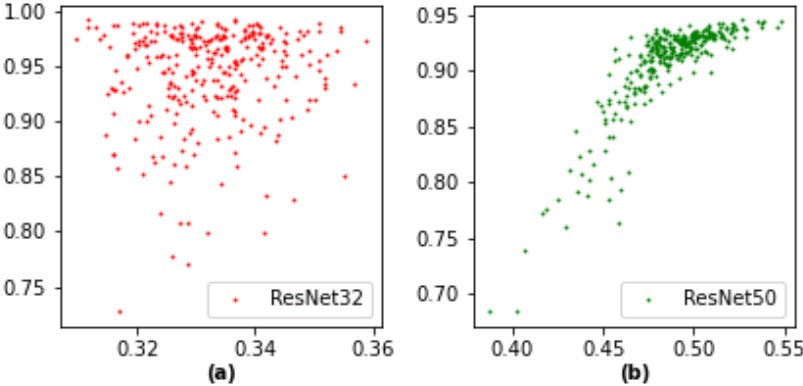

Figure 5: Angle contraction of random pairs in centered cifar10 dataset for ResNet32 and ResNet50

## 6 CONCLUSION

We find that the main source of the angle contraction happens at the global pooling layer in most deep learning models. We show that pooling strategies are data-insensitive in FCN because the pooling will reduce the variance drastically while keeping the expectation unchanged. This phenomenon still holds for max pooling in CNN but fails for average pooling in CNN, since the variance is not reduced due to the complex correlation between patches. We hope our work will shed light on the understanding of the deep learning black box and improving the performance of the learning models.

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

## A    PROOF OF THEOREM 4.1

*Proof.* Since average pooling and max pooling are positive homogeneous, without loss of generality, we assume that $\mathrm{Var}(x_i) = \mathrm{Var}(y_i) = 1$ for all $i \in [m]$. If we can show that $\|g(x) - g(y)\| \leq \epsilon$ with high probability for some small positive $\epsilon$ ($g$ is any function), the correlation between $g(x)$ and $g(y)$ is then lower bounded by

$$
\begin{aligned}
\rho(g(x), g(y)) &= \frac{\langle g(x), g(y) \rangle}{\|g(x)\| \|g(y)\|} \\
&= \frac{1}{2} \frac{\|g(x)\|^2 + \|g(y)\|^2 - \|g(x) - g(y)\|^2}{\|g(x)\| \|g(y)\|} \\
&= \frac{1}{2} \left( \frac{\|g(x)\|}{\|g(y)\|} + \frac{\|g(y)\|}{\|g(x)\|} \right) - \frac{1}{2} \frac{\|g(x) - g(y)\|^2}{\|g(x)\| \|g(y)\|} \\
&\geq 1 - \frac{1}{2} \frac{\|g(x) - g(y)\|^2}{\|g(x)\| \|g(y)\|} \\
&\geq 1 - \frac{1}{2} \left( \frac{\|g(x)\|}{\epsilon} - 1 \right)^{-2},
\end{aligned}
$$

which is close to 1. We want to show that $g = \mathcal{A}$ yields collapsed features. Indeed, for average pooling $\mathcal{A} : \mathbb{R}^{mn} \to \mathbb{R}^m$,

$$
\|\mathcal{A}(F(x)) - \mathcal{A}(F(y))\|^2 = \sum_{i=1}^{m} (\mathcal{A}(F(x))_i - \mathcal{A}(F(y))_i)^2 \tag{10}
$$

is a sum of i.i.d. random variables. In particular, for each $i \in [m]$, since $\mathcal{A}(F(x))_i$ and $\mathcal{A}(F(y))_i$ are identical distribution (not necessarily independent), $\mathbb{E}[\mathcal{A}(F(x))_i] = \mathbb{E}[\mathcal{A}(F(y))_i]$, and $\mathrm{Var}(\mathcal{A}(F(x))_i) = \mathrm{Var}(\mathcal{A}(F(y))_i) = (\frac{1}{2} - \frac{1}{2\pi}) \frac{1}{n} \leq \frac{1}{4n}$

$$
\begin{aligned}
&\mathbb{E}[(\mathcal{A}(F(x))_i - \mathcal{A}(F(y))_i)^2] \\
&= \mathbb{E}[(\mathcal{A}(F(x))_i - \mathbb{E}[\mathcal{A}(F(x))_i] - \mathcal{A}(F(x))_i + \mathbb{E}[\mathcal{A}(F(y))_i])]^2 \\
&\leq \mathrm{Var}(\mathcal{A}(F(x))_i) + \mathrm{Var}(\mathcal{A}(F(y))_i) \\
&= 2\mathrm{Var}(\mathcal{A}(F(x))_i) \\
&= \frac{1}{2n}
\end{aligned}
$$

and moreover,

$$
\begin{aligned}
&\|\mathcal{A}(F(x))_i - \mathcal{A}(F(y))_i\|_{\psi_2} \\
&\leq \|\mathcal{A}(F(x))_i - \mathbb{E}[\mathcal{A}(F(x))_i]\|_{\psi_2} + \|\mathcal{A}(F(y))_i - \mathbb{E}[\mathcal{A}(F(y))_i]\|_{\psi_2} \\
&\leq 2\|\mathcal{A}(F(y)(x))_i - \mathbb{E}[\mathcal{A}(F(x))_i]\|_{\psi_2} \\
&\leq 2\sqrt{\frac{C}{n}} \|x_i\|_{\psi_2} \\
&\leq 3\sqrt{\frac{C}{n}},
\end{aligned}
$$

where the first inequality holds as the triangle inequality of normed space and the third inequality holds since $\|\sum_{j=1}^{n} X_i\|_{\psi_2}^2 \leq C \sum_{j=1}^{n} \|X_i\|_{\psi_2}^2$ for independent mean-zero sub-gaussian $X_i$'s. The above calculation yields

$$
\|(\mathcal{A}(F(x))_i - \mathcal{A}(F(y))_i)^2\|_{\psi_1} = \|\mathcal{A}(F(x))_i - \mathcal{A}(F(y))_i\|_{\psi_2}^2 \leq \frac{9C}{n}.
$$

By Bernstein's inequality for sub-exponential random variables [vershynin 2018],

$$
\begin{aligned}
&\mathbf{P}\{\|\mathcal{A}(F(x)) - \mathcal{A}(F(y))\|^2 > \frac{10Cm}{n}\} \\
&\leq \mathbf{P}\{\|\mathcal{A}(F(x)) - \mathcal{A}(F(y))\|^2 - \mathbb{E}[(\mathcal{A}(F(x))_i - \mathcal{A}(F(y))_i)^2] > 9C\mathbb{E}[(\mathcal{A}(F(x))_i - \mathcal{A}(F(y))_i)^2]\} \\
&\leq \exp(-c \min(\frac{m^2}{4n}, \frac{m}{2})).
\end{aligned}
$$

Now it is left to estimate the value of $\|\mathcal{A}(F(x))\|$:

$$\|\mathcal{A}(F(x))\|^2$$

$$=\sum_{i=1}^{m}\mathcal{A}(F(x))_i^2$$

$$=\sum_{i=1}^{m}(\mathcal{A}(F(x))_i - \mathbb{E}[\mathcal{A}(F(x))_i] + \mathbb{E}[\mathcal{A}(F(x))_i])^2$$

$$=\sum_{i=1}^{m}(\mathcal{A}(F(x))_i - \mathbb{E}[\mathcal{A}(F(x))_i])^2 + m(\mathbb{E}[\mathcal{A}(F(x))_i])^2 - 2\mathbb{E}[\mathcal{A}(F(x))_i](\sum_{i=1}^{m}\mathcal{A}(F(x))_i - \mathbb{E}[\mathcal{A}(F(x))_i])$$

$$\geq \frac{m}{2}(\mathbb{E}[\mathcal{A}(F(x))_i])^2$$

$$\geq \frac{m}{15}$$

which is due to the Bernstein inequality and Hoeffding inequality with $\|(\mathcal{A}(F(x))_i - \mathbb{E}[\mathcal{A}(F(x))_i])^2\|_{\psi_1} \leq \frac{9C}{n}$ and $\|\mathcal{A}(F(x))_i - \mathbb{E}[\mathcal{A}(F(x))_i]\|_{\psi_2} \leq 3\sqrt{\frac{C}{n}}$ respectively Vershynin (2018). Applying the union bound, we conclude that the correlation is around $1 - O(n^{-1})$ with high probability. □

## B    PROOF OF THEOREM 4.2

*Proof.* For max pooling operator $\mathcal{M} : \mathbb{R}^{mn} \to \mathbb{R}^n$, note that $\max(r(x_1), \ldots, r(x_n)) = \max(0, x_1, \ldots, x_n)$, and Borell-TIS Inequality Adler (2007) for the maximum of independent standard normal random variables is sub-gaussian such that

$$\mathbf{P}\{|\mathcal{M}(x)_i - \mathbb{E}[\mathcal{M}(x)_i]| > t\} < 2\exp(-\frac{t^2}{2}) \tag{11}$$

for all $i \in [m]$ and $t > 0$. Equivalently, there exists a universal constant $K > 0$ such that $\|\mathcal{M}(F(x))_i - \mathbb{E}[\mathcal{M}(F(x))_i]\|_{\psi_2} \leq K$. Then the argument w.r.t $\mathcal{A}$ is applicable to $\mathcal{M}$ and we obtain with high probability

$$\|\mathcal{M}(F(x))_i - \mathcal{M}(F(y))_i\|_{\psi_2} \leq K\,; \tag{12}$$

$$\|\|\mathcal{M}(F(x)) - \mathcal{M}(F(y))\|^2\|_{\psi_1} \leq nK^2\,; \tag{13}$$

$$\mathbb{E}[(\mathcal{M}(F(x))_i - \mathcal{M}(F(y))_i)^2] \leq 2/\log(n)\,; \tag{14}$$

$$\|\|\mathcal{M}(F(x))\|^2\|_{\psi_1} \geq Cn\log n \tag{15}$$

The third inequality is from Talagrand's $L^1 - L^2$ Inequality Chatterjee (2014); the desired result of the theorem follows from Bernstein's Inequality again. □

The similarity of the two proofs is that the variance can be controlled very small while the expectation is relatively large.

## C    PROOF OF COROLLARY 4.3

*Proof.* we use the fact that $\rho(x, y) = \rho(ax, by)$ where $a$ and $b$ are positive constants. let $\gamma = \frac{\mathbb{E}\mathcal{M}_y}{\mathbb{E}\mathcal{M}_x}$ then $\rho(\gamma\mathcal{M}_x, \mathcal{M}_y) = \rho(\mathcal{M}_x, \mathcal{M}_y)$. Right now the $\mathbb{E}\gamma\mathcal{M}_x = \mathbb{E}\mathcal{M}_y$, and the technique of proving theorem 4.2 is applicable to the scene, thus the bound still holds. □

## D  UPPER BOUND OF THE VARIANCE OF AVERAGE POOLING FEATURE

*Proof.* Note that $\mathcal{A}r(W^s * x)$'s are i.i.d random variables, for any $w, v \in \mathbb{R}^{k \times k}$,

$$|\mathcal{A}r(w * x) - \mathcal{A}r(v * x)|$$

$$= \frac{1}{d^2} |\sum_{i,j}^{d^2} r(w * x)_{ij} - r(v * x)_{ij}|$$

$$\overset{(i)}{\leq} \frac{1}{d^2} \sum_{i,j=1}^{d} |((w - v) * x)_{ij}|$$

$$= \frac{1}{d^2} \sum_{i \in \mathcal{D}} \langle \alpha_i, w - v \rangle$$

$$\overset{(ii)}{\leq} \frac{\sum_{i \in \mathcal{D}} \|\alpha_i\|}{d^2} \|w - v\|$$

The inequality (i) holds since $r(\cdot)$ is 1-lipschitz; (ii) is Cauchy-Schwarz inequality. $\square$

By Gaussian Poincare's Inequality, $\text{Var}(\mathcal{A}r(w * x)) \leq (\frac{\sum_{i \in \mathcal{D}} \|\alpha_i\|}{d^2})^2$, which has the same order of magnitude of $\mathbb{E}[(\mathcal{A}r(w * x))^2]$. The experiment shows that average pooling has a large variability of angle contraction, which implies that the estimation of variance is tight without extra assumptions on the distribution of data. This indicates that average pooling has more intricate properties than max pooling.

## E  PROOF OF THE MEAN COSINE SIMILARITY OF AVERAGE POOLING IS CLOSE TO 1

*Proof.*

$$\mathcal{C} := \frac{\mathbb{E}[\mathcal{A}r(x)\mathcal{A}r(y)]}{(\mathbb{E}[(\mathcal{A}r(x))^2]\mathbb{E}[(\mathcal{A}r(y))^2])^{1/2}} \tag{16}$$

$$= \frac{\mathbb{E}_{x,y} \sum_{i,j \in \mathcal{D}} \mathbb{E} r(\langle \alpha_i, W \rangle) r(\langle \beta_i, W \rangle)/d^4}{\sqrt{\mathbb{E}_{x,y} \sum_{i,j \in \mathcal{D}} \mathbb{E} r(\langle \alpha_i, W \rangle) r(\langle \alpha_j, W \rangle)} \sqrt{\mathbb{E}_{x,y} \sum_{m,l \in \mathcal{D}} \mathbb{E} r(\langle \beta_k, W \rangle) r(\langle \beta_l, W \rangle)/d^4}} \tag{17}$$

$$= \frac{\mathbb{E}_{x,y} \sum_{i,j \in \mathcal{D}} \hat{r}(\langle \frac{\alpha_i}{\|\alpha_i\|}, \frac{\beta_j}{\|\beta_j\|} \rangle) \|\alpha_i\| \|\beta_j\|}{\sqrt{\mathbb{E}_{x,y} \sum_{i,j \in \mathcal{D}} \hat{r}(\langle \frac{\alpha_i}{\|\alpha_i\|}, \frac{\alpha_j}{\|\alpha_j\|} \rangle) \|\alpha_i\| \|\alpha_j\|} \sqrt{\mathbb{E}_{x,y} \sum_{i,j \in \mathcal{D}} \hat{r}(\langle \frac{\beta_m}{\|\beta_m\|}, \frac{\beta_l}{\|\beta_l\|} \rangle) \|\beta_m\| \|\beta_l\|}} \tag{18}$$

$$= \frac{\mathbb{E}_{x,y} \sum_{i,j \in \mathcal{D}} \hat{r}(\langle \frac{\alpha_i}{\|\alpha_i\|}, \frac{\beta_j}{\|\beta_j\|} \rangle) \|\alpha_i\| \|\beta_j\|}{\sqrt{\mathbb{E}_x \sum_{i,j \in \mathcal{D}} \hat{r}(\langle \frac{\alpha_i}{\|\alpha_i\|}, \frac{\alpha_j}{\|\alpha_j\|} \rangle) \|\alpha_i\| \|\alpha_j\|} \sqrt{\mathbb{E}_y \sum_{i,j \in \mathcal{D}} \hat{r}(\langle \frac{\beta_m}{\|\beta_m\|}, \frac{\beta_l}{\|\beta_l\|} \rangle) \|\beta_m\| \|\beta_l\|}}. \tag{19}$$

Since $x$ and $y$ are mutually independent, let $\Lambda = \mathbb{E}_{u,v} \hat{r}(\langle u, v \rangle)$, the nominator can be written as

$$\sum_{i,j \in \mathcal{D}} \mathbb{E}_{u,v} \hat{r}(\langle u, v \rangle) \mathbb{E}[s] \mathbb{E}[t] = \sum_{i,j \in \mathcal{D}} \mathbb{E}_{u,v} \hat{r}(\langle u, v \rangle) \mathbb{E}[s]^2 = d^4 \Lambda \mathbb{E}[s]^2, \tag{20}$$

where $u, v \overset{i.i.d}{\sim} Unif(\mathbb{S}^{r^2})$, $s, t \overset{i.i.d}{\sim} \chi_{k^2}$ are mutually independent due to the polar decomposition of standard normal random vector.

Two terms in the denominator are the same, thus

$$\sqrt{\mathbb{E}_x \sum_{i,j\in\mathcal{D}} \hat{r}(\langle \frac{\alpha_i}{\|\alpha_i\|}, \frac{\alpha_j}{\|\alpha_j\|}\rangle)\|\alpha_i\|\|\alpha_j\|} \sqrt{\mathbb{E}_y \sum_{i,j\in\mathcal{D}} \hat{r}(\langle \frac{\beta_m}{\|\beta_m\|}, \frac{\beta_l}{\|\beta_l\|}\rangle)\|\beta_m\|\|\beta_l\|} \qquad (21)$$

$$=\mathbb{E}_x \sum_{i,j\in\mathcal{D}} \hat{r}(\langle \frac{\alpha_i}{\|\alpha_i\|}, \frac{\alpha_j}{\|\alpha_j\|}\rangle)\|\alpha_i\|\|\alpha_j\|. \qquad (22)$$

Since $\alpha_i$ and $\alpha_j$ are not independent for some pairs of $i, j$, we decompose the sum over $j$ into two parts, the dependent part and the independent part:

$$\mathbb{E}_x \sum_{i\in\mathcal{D}} \Bigg( \sum_{\{j\in\mathcal{D}:ind(\alpha_j)\bigcap ind(\alpha_i)\neq\emptyset\}} \hat{r}(\langle \frac{\alpha_i}{\|\alpha_i\|}, \frac{\alpha_j}{\|\alpha_j\|}\rangle)\|\alpha_i\|\|\alpha_j\| \qquad (23)$$

$$+ \sum_{\{j\in\mathcal{D}:ind(\alpha_j)\bigcap ind(\alpha_i)=\emptyset\}} \hat{r}(\langle \frac{\alpha_i}{\|\alpha_i\|}, \frac{\alpha_j}{\|\alpha_j\|}\rangle)\|\alpha_i\|\|\alpha_j\| \Bigg) \qquad (24)$$

$$= d^2\mathbb{E}_x \Bigg( \underbrace{\sum_{\{j\in\mathcal{D}:ind(\alpha_j)\bigcap ind(\alpha_i)\neq\emptyset\}} \hat{r}(\langle \frac{\alpha_i}{\|\alpha_i\|}, \frac{\alpha_j}{\|\alpha_j\|}\rangle)\|\alpha_i\|\|\alpha_j\|}_{\textcircled{1}} \qquad (25)$$

$$+ \underbrace{\sum_{\{j\in\mathcal{D}:ind(\alpha_j)\bigcap ind(\alpha_i)=\emptyset\}} \hat{r}(\langle \frac{\alpha_i}{\|\alpha_i\|}, \frac{\alpha_j}{\|\alpha_j\|}\rangle)\|\alpha_i\|\|\alpha_j\|}_{\textcircled{2}} \Bigg). \qquad (26)$$

where we can show that

$$\textcircled{1} \overset{(i)}{\leq} d^2\mathbb{E}_x 9k^2\hat{r}(\langle \frac{\alpha_i}{\|\alpha_i\|}, \frac{\alpha_j}{\|\alpha_j\|}\rangle)\|\alpha_i\|\|\alpha_j\| \qquad (27)$$

$$\overset{(ii)}{\leq} (2k-1)^2 d^2\mathbb{E}_x \frac{1}{2}(\|\alpha_i\|^2 + \|\alpha_j\|^2) \qquad (28)$$

$$\overset{(iii)}{\leq} 2k^4 d^2, \qquad (29)$$

with $s' \sim \chi_{k^2}$. Inequality $(i)$ holds since for any patch $\alpha_i$ with size $k$, there are less or equal to $(2k-1)^2$'s patches that are dependent (i.e., not disjoint) with $\alpha$ (see the illustration in Figure F in the appendix); $(ii)$ follows from the fact that inequality $a^2 + b^2 \geq 2ab$. Moreover,

$$\textcircled{2} = (d^2 - 9k^2)d^2\Lambda E[s]^2 \qquad (30)$$

Thus we have

$$\mathcal{C} \geq \frac{d^4\Lambda E[s]^2}{2k^4 d^2 + (d^2 - 9k^2)d^2\Lambda E[s]^2} \qquad (31)$$

Assume $k = d^{\frac{1}{2}}$. Note $\Lambda \geq \hat{r}(E\langle u, v\rangle) = \hat{r}(0) = \frac{1}{\pi}$, and $E[s] \geq C$ for some universal constant $C$ Vershynin (2018), then $\mathcal{C} \geq (1 - \frac{4k^2}{d^2} + \frac{4\pi k^4}{d^2 E[s]^2})^{-1} \geq 1 - \frac{9}{C^2 d}$. The analysis can be extended to give a result of tail bound of the cosine similarity using standard technique similar to Theorem 4.1 due to the i.i.d nature of the kernels at initialization. $\qquad\square$

## F  THE ILLUSTRATION OF THE NON-DISJOINT PATCHES

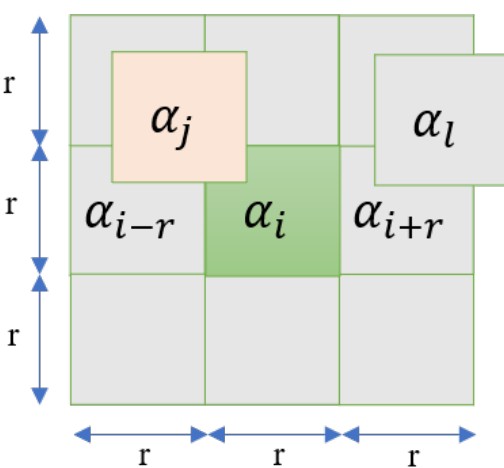

Figure 6: The patches that are disjoint or disjoint with the patch $\alpha_i$. In the picture $\alpha_j$ is not disjoint with $\alpha_i$, and other gray patches is disjoint with $\alpha_i$.

