# OpenReview forum: "The Contraction Property of Pooling Layer"
_ICLR.cc/2025/Conference — ICLR 2025 Conference Withdrawn Submission_

### Official Review · Reviewer_5h7o · 2024-10-29

**Soundness:** 2
**Presentation:** 2
**Contribution:** 1
**Rating:** 3
**Confidence:** 4

**Summary:**

The paper provides a theoretical analysis of pooling layers, specifically investigating the angle contraction behavior of various pooling strategies.

**Strengths:**

1- Although structural enhancements for generic CNNs are not a primary focus of the research community, theoretical analyses of various aspects of these models remain valuable. Such insights are still of interest and can contribute meaningfully to current trends in AI and deep learning research advancements.

2- The authors support their claims through theoretical analysis and proofs.

**Weaknesses:**

1- As I mentioned, the subject is quite old and unrelated to the research community's current trend.

2- The role of angle contraction and its impact on model performance remains unclear. What could be the potential significance of this study?

3- The figures should be more illustrative; the current ones are low in quality.

4- Only image-format inputs were used for the empirical evaluation, while other modalities (e.g., voice, time series) should also be included in the study.

5- What is the impact of multilayer networks on angle contraction? The study uses a global pooling layer for the CNN; what if additional layers are added and different pooling sizes are used?

**Questions:**

Please refer to the above section.

---

### Official Review · Reviewer_acTS · 2024-10-30

**Soundness:** 1
**Presentation:** 1
**Contribution:** 1
**Rating:** 1
**Confidence:** 4

**Summary:**

The authors claim that angle contraction occurs due to the pooling layer, and the cosine similarity between average pooling and max pooling demonstrates a difference in data dependency.

**Strengths:**

The mathematical notations are clearly stated.

**Weaknesses:**

This paper has several critical issues that need to be addressed to meet academic standards.

1. Limited Originality:  The main contribution of the paper has already been discussed in a previous reference.

2. Writing Quality: The paper has notable grammatical issues. Several expressions are grammatically incorrect, which disrupts the flow and clarity of the content. Improving sentence structure and word choice would significantly enhance the readability of the text.

3. Reference Formatting Errors: The citation style does not conform to proper academic referencing standards. For instance, the reference to "Poole et al. (2016)" is incorrectly formatted, which affects the readability and credibility of the document. Ensuring that all references follow a consistent format (e.g., APA, MLA) is essential for scholarly writing.

In summary, this paper requires substantial revisions to meet academic standards, particularly in reference formatting, writing clarity, and adherence to plagiarism policies.

**Questions:**

Why do you claim that your contribution is reporting the angle contraction phenomenon when it has already been discussed in Jakub and Nica's paper?

**Details Of Ethics Concerns:**

The first two sentences in the provided text appear to be directly copied from Cameron Jakub and Mihai Nica's work titled "Depth Degeneracy in Neural Networks: Vanishing Angles in Fully Connected ReLU Networks on Initialization." Academic integrity requires that direct quotes be properly cited with quotation marks and specific page numbers, or the ideas be paraphrased and appropriately credited to the original author. Failure to do so raises serious plagiarism concerns.

---

### Official Review · Reviewer_xSjc · 2024-11-03

**Soundness:** 3
**Presentation:** 3
**Contribution:** 2
**Rating:** 5
**Confidence:** 4

**Summary:**

The authors analyze angle contraction in convolutional neural networks from the angle of global pooling layers.
The authors conclude that these layers are significant contributors to this behavior and find differences between average pooling and max pooling in terms of how dataset-dependent the angle contraction is in CNNs. In contrast, the authors find that angle contraction FCNs is insensitive to the dataset.

**Strengths:**

- The authors shed new light into the well-established angle contraction from the perspective of pooling.
- The authors provide both experimental results and theoretical justification of the studied phenomenon.
- I commend that the authors were upfront about the limitations of their analysis.

**Weaknesses:**

- The analysis, while interesting, is limited and preliminary. I was expecting the analysis to include strided pooling layers that are frequently used for downsampling in typical CNNs, not only the penultimate layer.
- I miss a comparison with closely-related work by Voss, et al. (Distill 2021) where the authors report similar observations about global pooling.
- Some references are inaccurate and inappropriate for the cited context. For example the vanishing/exploding gradient problem was well reported by Hochreiter et al in the 1990s, long before the reference cited as [He et al 2015]. The same holds for expressive power of deep neural networks.
  - Voss, et al. "Visualizing weights." Distill 6.2 (2021): e00024-007.
  - Hochreiter, Sepp. "The vanishing gradient problem during learning recurrent neural nets and problem solutions." International Journal of Uncertainty, Fuzziness and Knowledge-Based Systems 6.02 (1998): 107-116.
  - Hochreiter, Sepp. "Untersuchungen zu dynamischen neuronalen Netzen." Diploma, Technische Universität München 91.1 (1991): 31.

There were numerous writing issues related to capitalization, articles, and citations:
- At the end of section 1.1, capitalize the first letter after every full stop.
- Please capitalize "we" in "". we call" (line 140) and  ". we observe" (line 325) .
- Please decapitalize "S" in "and m Since the two"
. "We abuse the symbols W" => please reword in a scientific manner.
- Please use \citep when the citation is not a natural part of the text and should be inside parentheses, and use \citet when the author names are part of the sentence with sound grammar.

**Questions:**

Voss, et al. also found the global pooling layers dictate the angles of the kernels in the penultimate layer.
How different are the conclusions present in this work?

Voss, et al. "Visualizing weights." Distill 6.2 (2021): e00024-007.

---

### Official Review · Reviewer_MWri · 2024-11-03

**Soundness:** 2
**Presentation:** 1
**Contribution:** 2
**Rating:** 3
**Confidence:** 3

**Summary:**

This paper studies angle contraction in pooling layers. It shows that angle contraction is not data-dependent in Fully Connected Networks (FCN) for average pooling and FCNs and CNNs for max pooling. It is, however, data-dependent in CNNs for average pooling. The paper performs both theoretical and empirical analysis on CIFAR10, CIFAR100 and MNIST datasets.

**Strengths:**

+ The paper studies angle contraction phenomenon in FCN and CNNs as a follow up of recent papers such as Cameron Jakub, et. al. `Vanishing Angles in Fully Connected ReLUs … `.
+ The papers attempts to explain the phenomenon both theoretically and empirically for a number of models and datasets (both real and generated)

**Weaknesses:**

+ Overall the study of contraction angles sounds inconclusive especially for the average pooling in different architectures. I’d recommend conducting more thorough experiments and studying the theory deeper for more grounded observations. At this point the discussion and experimentation sections seem slightly vague.
+ Figure 1: The paper reads: `... cosine similarity of relu-activated features … `, it however does not describe whether this cosine similarity is for different data points, layers or something else. There is a similar problem in the other plots as well. It would be better to describe whether we are talking about different data points here.
+ Figure 2: It looks like the figure is not cited in the paper and the description of the figure does not tell how to interpret it. It might be good to describe how the gaussian data is generated.
+ Figure 4 doesn’t seem to be referenced in the paper.
+ Figure 5: the paper is reference to figures 5 c-d but there are only a-b plots
The figure axes are not notated across all plots. Although the authors describe what the axes are, it would be better to annotate the axes and focus on the interpretation of the figure in the description of the plots.

**Minor**
+ Line 46: ue -> use ?
+ Line 165: 2 since-s in one sentence doesn't sound grammatically correct.

**Questions:**

+ When we see linear correlation between the cosine similarities (horizontal vs vertical axes), how do we interpret it from the angle contraction and data-dependence perspectives ? Is it possible to describe the experiments such as: plot XYZ shows angle contractions because of `W` and `U` and no data-dependence because of `E` and `P` ?
+ See weaknesses

---

### Note · Authors · 2024-12-07

I have read and agree with the venue's withdrawal policy on behalf of myself and my co-authors.